# Advancements in Artificial Intelligence-Based Decision Support Systems for Improving Construction Project Sustainability: A Systematic Literature Review

**Craig John Smith and Andy T. C. Wong \***

Department of Design Manufacturing and Engineering Management, University of Strathclyde, Glasgow G1 1XQ, UK; craig.smith@strath.ac.uk
**\*** Correspondence: andy.wong@strath.ac.uk

**Abstract:** This paper aims at evaluating the current state of research into artificial intelligence (AI)-based decision support systems (DSS) for improving construction project sustainability. The literature was systematically reviewed to explore the use of AI in the construction project lifecycle together with the consideration of the economic, environmental, and social goals of sustainability. A total of 2688 research papers were reviewed, and 77 papers were further analyzed, and the major tasks of the DSSs were categorized. Our review results suggest that the main research stream is dedicated to early-stage project prediction (50% of all papers), with artificial neural networks (ANNs) and fuzzy logic (FL) being the most popular AI algorithms in use. Hybrid AI models were used in 46% of all studies. The goal for economic sustainability is the most considered in research, with 87% of all papers considering this goal, and there is evidence given of a trend towards the environmental and social goals of sustainability receiving increasing attention throughout the latter half of the decade.

**Keywords:** decision support system; construction; artificial intelligence; machine learning; sustainability

## 1. Introduction

Project management in the construction sector has its own unique challenges, which have a clear impact on project success. These challenges relate to each project having variations in location, personnel, the equipment and logistics, as well as other factors such as economics and cost variations [1]. These can increase the degree of uncertainty during project planning and implementation, which can result in overspending, project delays, and disputes between the customer, employees, and contractors. In addition, traditional project management methods that are used by current construction companies rely heavily on the experience of project managers, while data are collected manually in a variety of non-digital formats through decentralized storage [2,3]. This leads to the use of delayed, flawed, or incomplete information during decision making, which jeopardizes process improvement. Relying heavily on empirical knowledge rather than a systematic approach often leads to wrong conclusions with substantial consequences.

To combat the above issues, research into the use of artificial intelligence (AI)-based decision support systems (DSS) has been gaining in popularity. AI technologies are becoming powerful tools throughout the world for improving project management; however, the advancement of construction management is still in its infancy and is adapting to the use of AI at a much slower pace than other sectors [4].

Moreover, the critical success factors (CSFs) of project management are changing alongside the technological advancements. In particular, the CSF evolution is obvious in construction project management due to key changes in societal needs at the start of this century. The goals of construction project management have moved on from focusing on cost and profit, scheduling, and quality to also consider other tangible and intangible

factors as well [5]. This trend motivates the measurement of project success with respect to the goals of economic, environmental, and social sustainability.

Here, the aim of this paper is to investigate the current literature for DSS technologies in construction project management with a focus on the use of AI for improving project sustainability. The two research questions (RQ1–2) that this paper intends to answer are:

RQ1: What are the trends in research for using AI in DSSs during the construction project lifecycle?

RQ2: What are the trends in relation to DSSs and construction project sustainability?

This paper is structured as follows: Section 2 will cover the background information related to the research, Section 3 will explain the method used for the study, and Section 4 will discuss the findings of the research. Suggestions for future research will then be explained in Section 5 before the conclusions of the study are provided in Section 6.

## 2. Background

To answer RQ1 and RQ2, it is important to define their key components, which cover an explanation of a DSS, the sustainability criteria, AI, and the construction project lifecycle. Sections 2.1 and 2.2 are extracted from our previous paper [6].

### 2.1. Decision Support Systems

A DSS can be defined as a computer-based aid, which is designed to assist project managers in decision making when the tasks at hand are of a complex nature [7,8]. Early DSSs could be defined as passive ones that would do what the users explicitly direct them to do, with a narrow range of decision-making capability [7]. In more recent years, the introduction of AI techniques has increased the capability of DSSs in construction. The basic structure of a DSS can be seen in Figure 1. This involves a user interface to support human-machine interaction, such as inputting data for analysis and receiving the guidance or recommendations in a human-readable format. The inference engine is the brain of the system, which utilizes AI algorithms to perform reasoning and computing. Hence, decisions or solutions to a problem can be derived based off the historical data stored in the knowledge base and the inputted data from the user. The knowledge base is where useful decision-making logics and historical data are stored to support the inference engine. The knowledge base will also be updated with new information/knowledge from interacting with users and resolving real-life issues. This will increase the sophistication of knowledge base and the intelligence of the system.

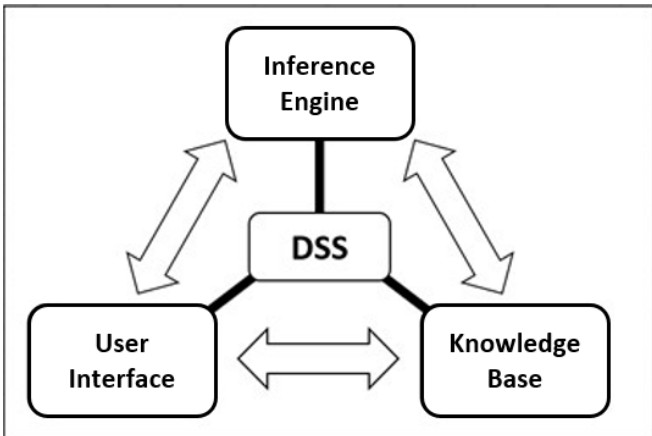

**Figure 1.** The three key components of a DSS: the user interface, the inference engine, and the knowledge database.

### 2.2. Sustainability in Construction

Project sustainability relies on the notion that the project should meet the present as well as future needs without compromising the others' interests. A widely accepted

view on sustainability is the idea of the three pillars of sustainability, which are defined as the economic, environmental, and social goals [9]. These are more informally defined as the consideration of the three Ps: profit, planet, and people [10]. The goals of each are explained as follows.

### 2.2.1. Economic Sustainability

The primary goal of economic sustainability is providing financial return compared to the utilized resources, adding value, and making financial profits while saving through minimizing expenditure [11]. Examples of this would include efficient project management, compliance with standards and regulations, and effective risk management and mitigation.

### 2.2.2. Environmental Sustainability

Looking further than the business alone, the goal of environmental sustainability is to minimize the negative impact of operations on the environment and to maintain and improve the environment. This includes reducing energy usage, limiting material usage, and utilizing environmentally friendly materials [12]. The goals of environmental sustainability have often gone together with economic sustainability, as the reduction of waste materials and compensating for inefficient working methods will reduce financial waste and increase profits.

### 2.2.3. Social Sustainability

The goal of social sustainability is to maintain and improve the quality of human life. This includes customers, employees, contractors, and all other parties who may be affected by the work carried out during a project. This can be through improving health and well-being, better training, and development; improving workplace diversity; and contributing to the betterment of society [13]. The benefits of social sustainability are the improvement of the morale and well-being of company personnel; improving relations with suppliers, customers, and the affected parties; as well as improving reputations both locally and internationally.

### 2.3. Artificial Intelligence

AI is "the theory and development of computer systems able to perform tasks normally requiring human intelligence, such as visual perception, speech recognition, decision-making, and translation between languages" [14]. This study will focus primarily on the applications of AI for decision making. Relevant types of AI to this study are provided in this section.

### 2.3.1. Machine Learning

This is the process of developing computer programs that learn from past data to make predictions without being explicitly programmed to do so. The learning methods consist of supervised data; learning from labelled datasets for both the input and desired result and unsupervised learning; for structuring unlabeled data and reinforcement learning (RL); and mapping from situations to actions to maximize the reward [15]. Examples of machine learning algorithms include multivariate-linear regression (MLR), logistic regression (LR), support vector machine (SVM), decision tree (DT), random forest (RF), K-means, Bayesian inference (BI), and artificial neural network (ANN).

### 2.3.2. Fuzzy Logic

In the real world, especially in project management, there are situations in which human reasoning is required for decision making, which may have a level of uncertainty for what the right choice may be. A tool to combat these situations is fuzzy logic (FL), first introduced in 1965 by Lotfi Zadeh [16]. This is a technique that can measure the degree of correctness of uncertain data, and it has been widely adopted in real-world systems to tackle ill-defined and complex problems that have incomplete and imprecise

information [17]. Rather than measure something to be true or false, fuzzy logic is used to quantify the level of truth. In the context of the articles researched in this study, fuzzy logic is used primarily for quantifying the knowledge of experts from ranked questionnaires [18], capturing human reasoning for various applications in decision making.

### 2.3.3. Natural Language Processing

Natural language is what we humans use to communicate information as opposed to computer programming languages, examples being English or Mandarin. In order for natural language to be interpreted by a computer, a tool known as natural language processing (NLP) is applied [19]. This is concerned with creating computational models that will resemble the linguistic capability of human beings. This includes reading, writing, listening, and speaking [20]. NLP is used to convert natural language into computer readable language for applications in social media, customer service, e-commerce, education, entertainment, finance, and healthcare [21]. In relation to construction project management, the processing of typed documentation and reporting can be computed, and knowledge can be gained through the use of NLP. Examples being the evaluation of accident reports in the construction sector for determining the precursors for accidents [22].

### 2.3.4. Evolutionary Algorithms

Evolutionary algorithms are a tool that relates biology, artificial intelligence, numerical optimization, and decision support for a diverse field of engineering applications. These algorithms utilize models based on organic evolution for intelligent optimization [23]. The task of intelligent optimization involves searching for the best result to minimize or maximize an objective function subjected to defined constraints [4]. An example of this type of algorithm would be the genetic algorithm (GA). This can be used for the optimization of the results of a system to improve model performance in decision making for construction project management [24].

### *2.4. Construction Project Lifecycle*

The construction industry can be defined as the construction, extension, installation, repair and maintenance, renewal, removal, alteration, dismantling or demolition of any building or structure, transport infrastructures, power, and water services [25]. This covers both commercial and industrial applications, residential buildings, as well as infrastructure items, such as transport routes, water service stations, and the connecting pipelines. Over this wide field of construction, projects can be categorized into five stages. These stages are initiation, planning, execution, controlling, and closing [26]. A flowchart explaining each stage of the project lifecycle is given in Figure 2.

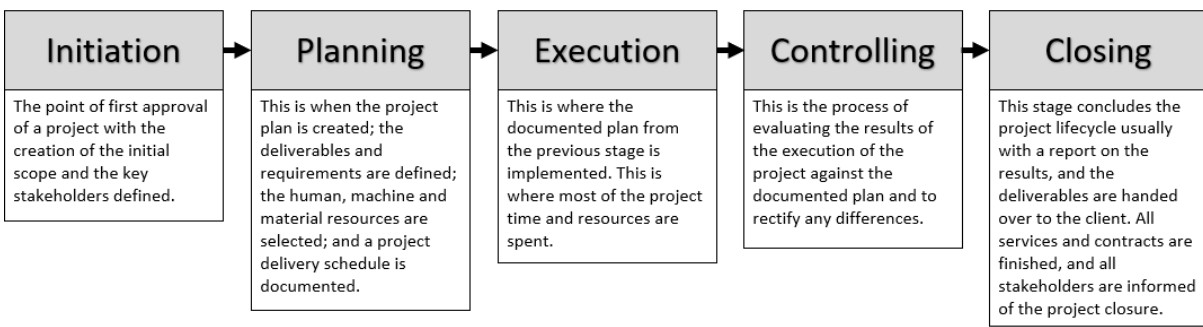

**Figure 2.** A flowchart of the five stages of the project lifecycle.

The studies that will be discussed in this article will be analyzed through the lens of the categorization shown in Figure 2, with the application of research defined in one or more of these five areas.

## 3. Literature Review Methodology

The aim of a systematic approach to reviewing literature is to identify all the empirical evidence within a pre-specified inclusion criteria to answer a particular research hypothesis [27]. The nature of this method reduces subjectivity in the research, leading to a reduction in bias. This method also allows for a quantitative analysis of papers to determine overall trends and relationships within a study. This literature review was conducted with a systematic approach, a flow diagram of the process is shown in Figure 3. The three key stages of selection are identification, screening, and assessment.

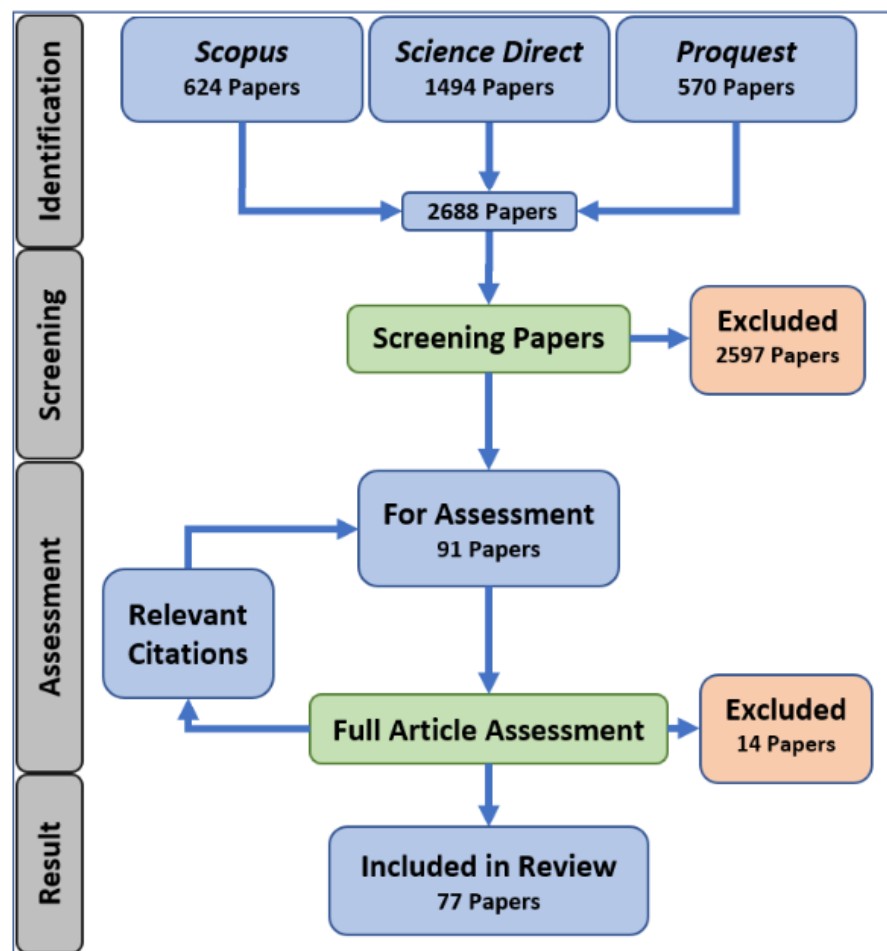

**Figure 3.** The flow diagram for the method of paper selection.

### 3.1. Paper Identification

This review is to investigate the current state of AI-based DSS techniques in the construction sector to improve project sustainability. The key words, thus, were decision support system, construction, project sustainability, artificial intelligence, and machine learning (ML) for literature searching within the three well-known databases: *Scopus, Science Direct,* and *ProQuest*. The search with the stated key words was conducted on the article titles, keywords, and abstracts. Our search was limited to papers written only in English between 2010 to present and peer-reviewed articles only. The relevant disciplines were Engineering; Computer Science; Mathematics; Business, Management, and Accounting; and Decision Sciences. This resulted in a total of 624, 1494, and 570 papers from *Scopus, Science Direct,* and *ProQuest,* respectively, for a grand total of 2688 papers to screen.

*3.2. Screening*

For screening, the abstracts were read for each of the identified papers. As there is a variety of frameworks that can be defined as a DSS [28], papers were deemed as relevant if DSSs or decision making were examined. Among all relevant papers, two levels were defined to differentiate relevance among them. The top level of relevance focused on papers that included research adopting AI with sustainability goals of construction projects. The second level of relevance included papers investigating any two of the three of adopting AI, sustainability goals, and construction projects for decision making and DSSs. These two levels of significance were used for the screening the papers. If papers achieved either of the two stated levels of relevance in the abstract, they would be included for full-article assessment. If these levels of relevance were not achieved, then the papers would be eliminated from the study. This resulted in 91 papers selected for the next stage of the review, which is the full-article assessment.

*3.3. Assessment*

The full content of each of the 91 remaining papers was assessed with respect to the criteria shown in Table 1, using a similar approach to [29]. A further 14 papers were eliminated from the review at this stage; hence, only 77 papers remained, which included 9 literature reviews and 68 research papers for in-depth analysis. These papers were analyzed, and the advancements in the field are discussed in the next section.

**Table 1.** A table describing the assessment criteria for this literature review.

| Data to Collect | Description |
|---|---|
| Task | Intended task of the DSS |
| Title | Title of the paper |
| Author (s) | List of authors |
| Contribution | Contribution to literature |
| Limitations | Potential improvements |
| Year | Year of publication |
| AI | Type of AI algorithm used |
| Sustainability | Economic, environmental, or social goal considerations |
| Stage of Construction | Lifecycle stage of operation |
| Institution | Location of the institution that carried out the study |
| Case study | Where is the case study located |

## 4. Review Findings

This section describes the findings of this research. This will start with the categorization of the papers by the task of the DSSs in all assessed papers. This will then be followed by the findings related to AI, sustainability, and the project lifecycle.

*4.1. Categorizing the Task of the DSSs*

The areas in which a DSS may be applied in the construction project lifecycle varies in the forms of data being used, the tasks of the inference system, and when in the cycle these tools may be applied. The 68 papers listed in this study have been organized into six distinct categories based on the task of the DSS. There is early-stage project prediction (EPP), which takes up 50% of all studies, with sub-categories focusing on various metrics for performance measurement; there is dynamic performance prediction (DPP), which takes up 17% of all studies; and then, there are papers focused on contractor and supplier evaluation, site logistics, design optimization, and safety risk assessments (SRA). Looking at Figure 4, there has been an increase in studies over the second half of the last decade, which would suggest an increase in interest in this field. It can also be noticed that the EPP has a near-consistent level of interest throughout the decade with other areas such as SRA, site logistics, DPP, design optimization, and supplier evaluation having more studies from

2016 onward. This shows a growth in the quantity of studies over this period but also a growth through increased variety of application.

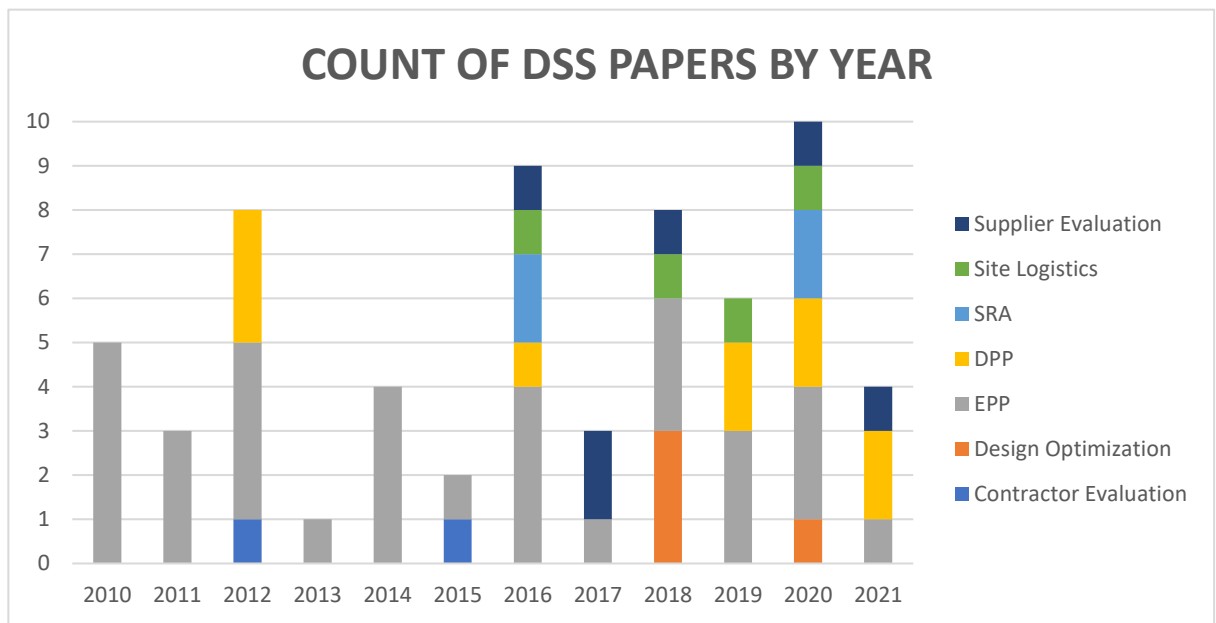

**Figure 4.** The count of each DSS category examined in papers by year of publication.

Each of the chosen categories of DSS application were analyzed against the type of AI used, the considerations for sustainability, and what stage in the construction lifecycle the system operates. The findings can be seen in Table 2. The following sub-sections will discuss the categories in more detail, followed by discussion on the use of AI, sustainability, and the project lifecycle.

**Table 2.** A table of the count of types of AI, sustainability criteria, and the stage of construction against each category of DSS.

| DSS Category | | Type of AI Used | | | | | | | | | | | | Sustainability Crit. | | | Stage of Lifecycle | | | | | |
|---|---|---|---|---|---|---|---|---|---|---|---|---|---|---|---|---|---|---|---|---|---|---|
| Category Name | Example References | ANN | BI | CBR | CNN | DT | FL | GA | GD | MLR | RF | RL | SVM | Eco | Env | Soc | Init. | Plan. | Exec. | Cont. | Close | All |
| Contractor and Supplier Evaluation | [18,30,31] | | | | | | 6 | | | | | | | 7 | 5 | 6 | 1 | 1 | | | | |
| Design Optimization | [32,33] | 1 | | | | | | | | | | 1 | | 4 | 4 | 2 | | 4 | | | | |
| Early-stage Project Prediction | [34–38] | 19 | | 6 | | 1 | 11 | 6 | | 6 | 1 | | 4 | 33 | 7 | 8 | 11 | 22 | | | | |
| Dynamic Performance Prediction | [39–41] | 3 | 1 | 1 | | 2 | 2 | 1 | | | 1 | | 2 | 10 | 2 | 3 | | | 1 | | | 8 |
| Safety Risk Assessment | [42,43] | 2 | | | 1 | | 1 | | 1 | | 1 | | 1 | | | 4 | | | 4 | | | |
| Site Logistics | [44–46] | | | | | | 1 | 1 | | | | | | 4 | 3 | 3 | 2 | 2 | | | | |
| Total | | 25 | 1 | 7 | 1 | 3 | 21 | 8 | 1 | 6 | 3 | 1 | 7 | 58 | 21 | 26 | 12 | 29 | 7 | | | 8 |

### 4.1.1. Contractor and Supplier Evaluation

An area of research where DSSs have been applied with sustainability criteria using AI is for evaluating suppliers. Ref. [47] refined sustainability criteria for the selection of suppliers with the assistance of academics and industry experts for defining the importance of applicability of criteria taken from the literature. Fuzzy preference programming was then used to allocate weights to each of the sustainability criteria, resulting in graded levels of importance reducing from economic to environmental to social criteria. Fuzzy Technique for Order of Preference by Similarity to Ideal Solution (FTOPSIS) was then used for supplier selection. Others have also used a similar approach, with [48] also using fuzzy logic for weight definition but using VIKOR for the selection of the projects. Another example is [49], who used VIKOR for project selection but with analytical hierarchy processing

for the weight definition. The sustainable selection criteria in all these studies are great examples of a drive towards sustainability, and these are just some examples of the few studies into supplier selection in manufacturing [48]. It can be seen in these studies that all the evaluation criteria are defined through subjective opinions of experts related to the work, and there is a lack of quantitative data. Combining these two data types may prove beneficial for supplier selections. These are all focused on the manufacturing industry, which does have a different format from the construction industry for supplier selection and would have differences in the selection criteria based of the unique aspects of construction projects when compared to manufacturing.

A couple of studies were found that applied a similar approach to the supplier selection, for example, ref. [50] created a framework software platform for the selection of construction project sub-contractors using fuzzy logic algorithms. This is done through providing linguistic ranking and marking questionnaires to experts at multiple stages in the selection process; then, fuzzy set theory is used to group, quantify, and rank sub-contractors. This is focused on international construction projects and is limited in that it does not factor in the built working relationships between contractors and sub-contractors. Additionally, all the data are provided based off expert opinion, so there is a level of subjectivity in the process of selection. Some factors that have not been studied in the recorded papers is the evaluation of contractors throughout the execution stage of the project lifecycle and the use of more complex AI models for tasks in addition to quantifying linguistic data.

### 4.1.2. Design Optimization

This category focuses on the use of DSSs for improving design in construction. All papers related to this category were published from 2018 onwards, and it appears that sustainable design is the main driver for all of them. All papers related to design optimization considered the economic and environmental goals of sustainability, while only half of them considered social sustainability.

Ref. [32] highlighted ongoing research into the use of DSSs for sustainable building material selection in the design stages, with a key focus on incorporating criteria for the environmental goals of sustainability. Ref. [51] developed a DSS for helping design engineers to choose sustainable materials during the planning stage of construction for pavement design. This method not only considers economic, environmental, and social goals during the project lifecycle but also for the maintenance of the materials during the lifecycle of the product. An example of AI being used for DSSs for design optimization would be [52], who developed a DSS for concept-design decision making in the construction industry. They adopted a Markov decision process (MDP) and RL for this DSS. The aim of this model was to implement value engineering from the manufacturing section into the construction design phase. The focus was to achieve optimization against environmental, economic, and social criteria. Utilizing the MDP approach was especially useful, as the structure of this approach has similarities to the decision-making system that engineers manually carry out in the concept-design stage of construction projects. The method was tested using the concept design of a house, and the design was optimized, which showed a positive result; however, there is area for improvement by adding feedback complexity and representing the interdependencies between different decisions at different stages of design.

### 4.1.3. Early-Stage Project Predictions

The most popular application for a DSS from the last ten years is for making predictions of project performance at the initiation and planning stages of the project lifecycle. This can be for project cost prediction, project delays, and for risk in project selection. These areas of study all follow the same approach of utilizing historical project performance data and key parameters to train an algorithm for predicting the resultant performance given the same input parameters for a new test project. This is an especially useful tool, as it provides the project manager with a quantifiable method for selecting which projects to

choose during the initiation stage of the lifecycle or how best to plan for a project prior to execution. The most popular algorithms to be used are ANNs and more recent models, which include hybrids with FL for quantifying qualitative data and genetic algorithms (GA) for optimizing the weights of the parameters [24,53–55]. Case-based reasoning (CBR) has also been studied, utilizing previous similar cases of projects to make predictions [56–60]. It can be observed in Table 2 that most of the research into project predictions examine the economic pillar of sustainability with 76% of all EPP research solely focusing on the economic sustainability goals. Research considering environmental and social goals is the minority, equating to approximately 25% of studies.

The ability to predict the cost of a project accurately has a significant impact on the economic sustainability of a construction project. This could help to ensure project success for choosing which project, equipment, or contractors to use or for determining the number of resources to provide. In construction engineering management, cost estimation at the start of a project is key to preventing cost overruns and ensuring project success [61,62].

For improving the accuracy of predictions, ref. [63] employed the use of an ANN to improve the prediction accuracy of water and sewer service project, as there were discrepancies of 60% error in predictions from standard practice in project cost estimation. Utilizing this ANN, they managed to reduce the error down to 20%. This is a clear improvement although this level of inaccuracy is still high when compared to other studies in the construction industry, an example being [64], who created a model for predicting the cost of building construction projects in Nigeria, also using an ANN model for cost prediction. Based off refined input parameters from 243 questionnaires given to experts in the field and an ANN with two hidden layers and sigmoid transfer functions, a high prediction accuracy was achieved having the mean absolute percentage error of only 5%. However, such high accuracy might be a result of high similarity over different building projects. In other words, the robustness of this model had not been tested on other building types, and hence, the generalization of the cost model is deemed low.

The majority of recent research into EPP has used hybrid AI models (58% of all EPP studies). Ref. [65] combined an ANN with FL to create an adaptive neuro-fuzzy inference system (ANFIS) for making cost predictions alongside principal items ration estimation method (PIREM) for keeping accuracy with fluctuating market prices. This method managed to achieve a mean prediction accuracy of within 10% of the actual cost when tested on residential building constructions in China. Another hybrid ANN model is defined as the evolutionary fuzzy hybrid neural network (EFHNN), which is a high-order neural network hybrid that utilized fuzzy inference for dealing with project uncertainties and a GA for optimizing the prediction accuracy. This model was tested on 28 building projects and compared to a singular linear ANN with increased accuracy in predicting the overall cost of projects and cost per internal categories of expenditure.

Another method for cost prediction is the use of case-based reasoning (CBR). This is an experience-based solution relating previously successful solutions to similar problems that occur in the future. Ref. [57] presented a case-based method for predicting construction costs using sports field installations as a case study. This did prove to have a mean absolute percentage error of 5%, and the method is not computationally intensive; however, it is limited by the number and type of previous cases as well as the similarity of the new projects. The model is validated with the construction of sports fields, which has highly similar tasks. Applying this model independently on more complex construction projects would better measure its robustness. Another CBR-based prediction model found in [56] compared the CBR method with hybrid models of CBR+ANN and CBR+FL, with the CBR+FL model proving most accurate with an average prediction error of 9% for predicting the cost of pump station projects. Leading further evidence towards the benefits of using hybrid AI solutions.

As well as determining the project cost at initial stages, there is benefit from predicting performance against other metrics. Project delays can have a substantial impact on success; ref. [66] created a method of categorizing project delays in the construction sector by use of

a random forest classifier with a genetic algorithm for result optimization. This method split projects into three categories of delay: less than 50% overrun, 50–100% overrun, and greater than 100% overrun. This model proved to have a classification accuracy of 91.67% and was deemed better the random forest model on its own, again highlighting the advantage of hybrid models for prediction. Additionally, the range of classification is substantial, and an improvement on the classification metric may be of larger benefit, as not all sectors of construction will find it acceptable to have ranges of 50% of the total specified time for classification. Although the construction industry is known for delays in projects, there is surprisingly little research into the use of AI technologies for predicting project delay likelihood at least not to the same level of depth as project cost prediction.

From all the previously stated studies into early prediction, environmental and social parameters and goals were not considered in the estimations. When it comes to predicting project risk for project selection, sustainability criteria have been a topic of research. The research presented by [67] introduced a method for selecting sustainability criteria for project selection and then used fuzzy preference programming for attribute weight selection and FL for aiding in the selection of projects. Fifteen different attributes, with five for each pillar, were selected from studying the literature and evaluation by three experts in construction engineering. Each of these attributes were given local (per attribute) and global (per category) weightings of importance and developed into 25 fuzzy rules within the system for defining the best alternative project to select. The system was evaluated using six projects from a construction company in Iran and compared with five other defuzzification methods and checked with a consistency index. Another study by [68] pulls a larger area of expertise with input from 53 experts in the form of a questionnaire. The weights and rules are built through analytical hierarchy processing and the novel rough set theory, respectively. This was tested on classifying 26 projects against sustainability criteria with a prediction accuracy ranging from 84–95%. These studies show real promise for the use of sustainability criteria in project selection although the weighting criteria is based largely off the subjective opinions of the experts, and the testing is based off a small quantity of projects. Considering quantitative data alongside the qualitative data may be something that could improve the robustness of the predictions, it would also be advantageous to apply the selection criteria to a larger program of projects for evaluation.

For all the reviewed studies, it appears that accuracy is as much dependent on the area of application and available data as it is for the algorithms used. The benefits of hybrid AI models are clear for improving prediction accuracies and for improving the robustness of models. Although the primary goal for project prediction is on cost estimation, other areas such as project risk are being investigated, which consider the social and environmental goals of sustainability as well as the economic.

### 4.1.4. Dynamic Performance Prediction

The main limitations in the EPP papers are that once a project begins the execution stage, there is usually a great deal of uncertainty, which can affect the predictive capability regardless of how powerful the AI algorithm is or the completeness of the pre-execution data. This is due to the fluctuating nature of the time dependent variables in construction project management, such as internal factors related to human resources during project execution or external factors, such as the impact of the weather on progress. Over the last 6 years only a minority of studies have dynamically predicted performance throughout the execution and all other stages of the life cycle. This allows for project managers to make educated decisions at the planning stages and then proactively improve project performance at the execution stage through to the controls and closure.

The authors of [24], who proposed the evolutionary fuzzy hybrid neural network (EFHNN) for project cost prediction at early stages, clearly understood the benefits of creating a dynamic performance-prediction tool. This hybrid is a combination of FL for dealing with uncertain data, a high-order ANN for making predictions, and GA for optimizing the results. The same authors published a paper on their dynamic prediction

performance method [69], which used the same hybrid AI algorithms to classify the performance of projects throughout the lifecycle. This classified project performance into four levels ranging from successful to disastrous, with inputs related to 10 time dependent variables, including change order data, weather impact, owner commitments, contractor commitments, recorded incidents, and overtime work. This model is classified with a high accuracy; however, the method was only validated against the highly similar evolutionary fuzzy neural inference model (EFNIM) and with only 12 projects for training and 3 projects for testing. This work could be taken further by comparing the model with a larger pool of AI models and with a much larger dataset.

A DSS framework presented in [2] combines the use of a manufacturing enterprise resource planning (ERP) system with building information modelling (BIM) for the purpose of guidance on project management, materials management, financial management, and human resource management, which will optimize project processes with the use of machine learning algorithms in the execution and control stages of a project. This is just a framework now, but this has the potential for real value in the future. Further study that includes the application of this system and the evaluation of AI models for project optimization is needed to gauge the overall effectiveness. When considering DPPs that consider sustainability criteria, [70] presented a framework for a sustainable construction project management index for evaluating construction projects. Six dimensions are defined: financial, scheduling, quality, safety, as well as informatization and "greenization". It is positive to see that research into dynamic construction performance measurement is being considered through the lens of sustainability. It is also key to note that from all studies into the DPP category, there are studies that have utilized AI and hybrids for improving the economic sustainability of projects [39], and there have been DPP studies that have considered all three goals of sustainability, but the use of AI models has not yet been seen to improve all three goals of sustainability in a single study of DPP.

### 4.1.5. Safety Risk Assessment

In the application of improving safety through the project lifecycle, there have been studies into the use of natural language processing (NLP) for analyzing injury reports. Ref. [43] used NLP to structure data from accident reports into attributes of incidents and the safety outcomes and then used random forest and stochastic gradient tree boosting for prediction. The models were able to have better predictive capability for defining the injury type, energy type involved, and the injured body part with a higher likelihood than at random, which gives evidence to the use of quantitative and empirical methods for evaluating safety compared to that of expert opinion and subjective judgment. One of the authors then took the study further, such as [22], which introduced a method for automatically determining valid accident precursors for accidents in the oil and gas sector. Three different ML techniques were used and compared. These are convolutional neural network (CNN), hierarchical attention network (HAN), and term frequency-inverse document frequency representation with support vector machine TF-IDF-SVM, which were used for NLP. All predictions of precursors performed better than random selection, and the TF-IDF + SVM method proved to be the most accurate. The data collected for these reports were quantitative in nature, and circumstantial and environmental information that contribute to hazards in the workplace were not considered.

### 4.1.6. Site Logistics

Site logistics can be defined as the control of the movement of people, equipment, and materials related to a work site. In this paper, the category for site logistics covers all the DSSs, which focus on improving site logistics with the use of AI and sustainability criteria. Ref. [45] developed a digital twin and DSS, which applied heuristic optimization and clustering for the purposes of silo replenishment on various construction sites during project execution. The purpose of this software tool is to predict the best routes for resupply vehicles to optimize vehicle usage and minimize work site stoppage times. Over a 3-year

period, this reduced logistic costs by up to 25% and with every kilometer of transport saved having a positive impact on the development of $CO_2$ emissions. The complexity of the digital twin and refill truck cost had a large effect on the cost reduction, which has left area for improvement.

Another study into improving logistics is [71], which covered the material transport routes, the emission levels and size of vehicles as well as the use of a construction consolidation center for minimizing the economic, social, and environmental impact of projects in Luxembourg. The multiple DSSs were tested on one large project, producing 47 alternative combinations of the above variables, with 5 reducing emissions and cost. This would be especially useful for projects in densely populated areas. Although this system considers sustainability in construction, it is limited in that it relies on experience of experts and mathematics. Using AI instead for optimization would provide a much larger pool of alternatives to consider for optimization. Most site logistics studies have considered all three pillars of sustainability, but there is room for further study into the benefits of AI for optimization considering hybrids to improve model performance.

### 4.2. Observations and Trends Related to AI

From Table 2, there is a wide range of AI models used in the literature, these various algorithms are listed in Table 3 with the associated abbreviations.

**Table 3.** A table expressing the various artificial intelligence algorithms from the study of the literature with the associated abbreviations.

| Abbreviation | Algorithm Title |
| --- | --- |
| ANN | Artificial Neural Network |
| CNN | Convolutional Neural Network |
| BI | Bayesian Inference |
| FL | Fuzzy Logic |
| CBR | Case Based Reasoning |
| DT | Decision Tree |
| RF | Random Forest |
| GA | Genetic Algorithm |
| GD | Gradient Descent |
| MLR | Multivariate Linear Regression |
| RL | Reinforced Learning |
| SVM | Support Vector Machine |

The overall trend in AI shows that complex prediction in the form of ANN and quantifying expert opinion using FL have had the most focus, which covered 37% and 31% of papers, respectively. GA is also popular for the optimization of models, with 12% of studies considering this. CBR, which focuses on the use of previous cases to advise project managers on how to progress in future projects, and two other machine learning algorithms, namely MLR and SVM, have been involved in approximately 10% of all studies. In total, 46% of the studies used hybrids of multiple AI algorithms; this was not an increasing trend, though. As the quantity of papers increased over the decade, the ratio of hybrid models decreased. This reduction in the ratio of hybrid models does coincide with the increase in studies for other applications of DSSs than EPP. In all, 50% of all studies are focused on EPP, which can lead to a bias of the overall results towards the EPP research.

Contractor and supplier evaluation papers have used only FL for quantifying expert opinion [47,50] while overlooking the potential of using machine learning techniques to examine empirical data alongside the opinion of experts. Design optimization and site logistics have only a few studies that use AI, and there is no obvious trend, but these two categories were published within the last six years, suggesting an increase in interest; hence, the potential of AI has not been fully explored in these areas. EPP is the most popular field of study and has been investigated with a wide variety of AI models. The most popular are the ANN and FL, with CBR, MLR, and GA also used in a considerable number of studies.

DPP has also been involved a wide variety of AI models, the most common being ANN, followed by DT, FL, and SVM. The similarity between AI used for the EPP and the DPP makes sense, as the tasks required for both are highly similar. They need project data, which can be empirical or linguistic from previous experience, and a value or values need to be predicted from the data. The main difference between those two categories is the stage of the construction lifecycle in which they operate. EPP operates only in the initiation and planning stages of the project lifecycle, while the DPP studies operate in the execution and all areas of the project lifecycle. The SRA papers focus on the use of NLP for analyzing accident reports, with one paper [22] comparing multiple AI algorithms for interpreting the data from risk assessments. These studies base the analysis purely off the wording in the incident reports but fail to address the circumstantial data that accompanies them. All the SRA papers were published from 2016, which suggests an increase in interest. These findings show that there is a wide variety of AI models used for construction DSS research. The type of AI used in research depends on the application of the DSS; however, some areas have been explored thoroughly, such as EPP, while site logistics, design optimization, and contractor and supplier evaluation have very few studies that explore the benefits of AI.

### 4.3. Observations and Trends Related to Sustainability

The counts shown in Table 2 for the sustainability criteria are defined as follows. If a paper considered factors related to each of the three pillars of sustainability, then accordingly, the count would be increased in "Eco", "Env", and "Soc" for the economic, environmental, and social goals, respectively. In total, 87% of studies considered the economic goal of sustainability. When looking at the other two goals of sustainability, 30% and 38% of studies included criteria from the environmental and social goals, respectively. As the societal switch to the consideration of environmental and social goals is recent, this is a reasonable result. A key factor to consider is the timeline for papers published over the last decade. From Figure 5, there is not only an increase in the number of papers published, but there is also a significant increase in the ratio of social and environmental goals being considered in research. It must be noted that this coincides with the increase in papers focused on design, site logistics, safety, and both supplier and contractor evaluation as shown earlier in Figure 4. These studies have been noted to have a high percentage of consideration for the environmental and social goals of sustainability. EPP primarily focuses on the economic goal of sustainability when viewing bidding, claims, and cost prediction; however, there has been an area of EPP focused on project risk, of which most studies consider the three sustainability goals [72,73]. This field of study has increased in regularity in the second half of the decade. DPP has the smallest ratio of consideration for goals other than the economic goal of sustainability; however, there were a couple of studies in 2019 and 2020 that adopted an approach towards all goals of sustainability [70,74]. These findings show that there is a shift towards research into project sustainability, with an increase in studies specifically aimed at improving sustainability in the latter half of the decade.

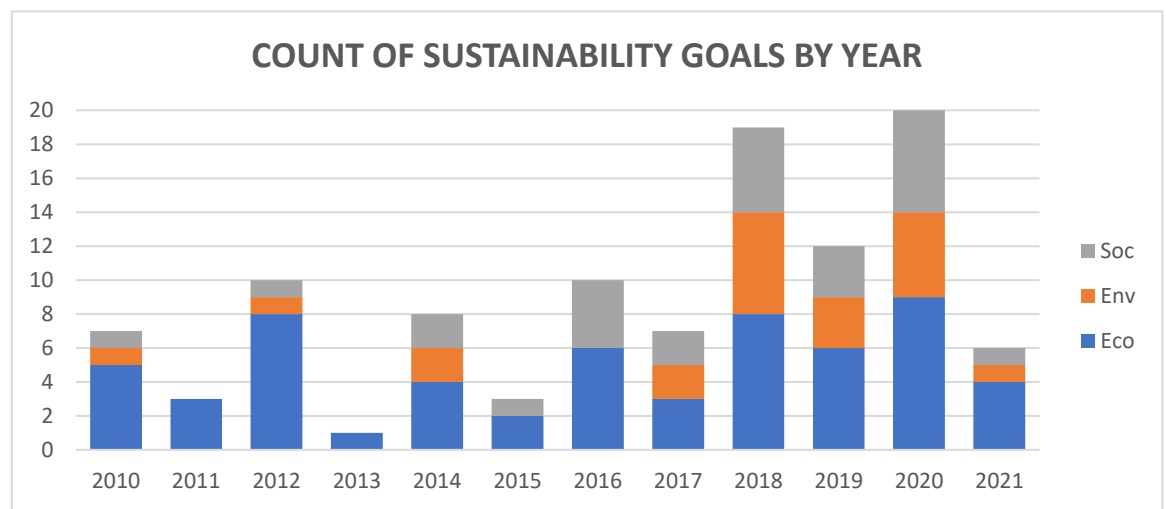

**Figure 5.** The count of each of the three goals of sustainability against the year the paper was published.

### 4.4. Observations and Trends Related to Project Lifecycle

Each of the papers were evaluated for the stage of project in which the operations of the decision tools were focused. The stages, as stated in Table 2, are initiation (init.), planning (plan.), execution (exec.), controls (cont.), and close. There are also studies that focus on the whole project lifecycle rather than any single stage. No study was recorded as solely operating in the controls or closing stages. Nearly half of all papers, 43% studied, were focused on decision support in the planning section, and a further 18% were focused on the initiation stage of construction. It can also be seen in Figure 6 that there has been a consistent production of publications focused on the planning and initiation stages of the project lifecycle. Research focused on the execution stage of the lifecycle has gained attention from 2016 onwards. The same trend is seen with studies that cover all five stages of the lifecycle although there was a spike of four papers in 2012 that covered all stages.

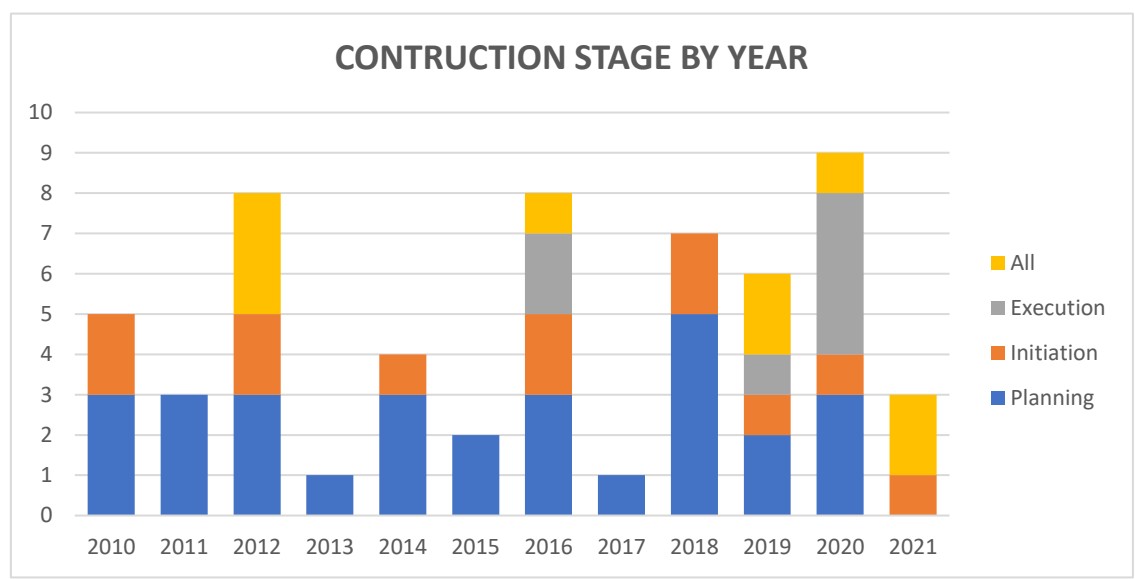

**Figure 6.** The count of projects focused on the stages of the project lifecycle against the year.

Analyzing by category, EPP focuses on making predictions at the early stages of the construction project lifecycle prior to the execution of the project plan. This is separate from DPP, which has similar characteristics but operates through the whole project lifecycle.

Contractor evaluation is also only researched at the beginning of the project lifecycle in all selected papers [18,50] despite contractors operating through the project execution. The design-optimization papers also only focus on the planning stage, but this is understandable; excluding reworks, all design is completed in the planning stage of the construction project lifecycle. The SRA studies understandably focus on the execution stage of the project lifecycle, as this is where the largest risks to health lie. Although the studies that focus on supplier evaluation are relevant to this research, as they include the use of AI for decision making and consider sustainability criteria, the supplier evaluation papers were focused on manufacturing projects, so these studies do not fit into the construction project lifecycle and were not considered for this stage of the analysis.

## 5. Future Research

Regarding RQ1, the observations made in this study suggest that there will be an increase in studies into DSS technology in the future, with AI being utilized for applications that cover all stages of the project lifecycle and for applications in management, logistics, and design. Regarding RQ2, although the economic goal of sustainability has had the most interest, there is a clear rise in studies that investigate the social and environmental goals through all applications of DSS technology. This suggests that there will be further studies considering all three pillars of sustainability in the future over all areas of the project lifecycle. Looking at more specific examples of potential study, the following sections will highlight future avenues of research by application of DSS.

### 5.1. Contractor Evaluation

Contractor evaluation was only ever considered at the point of selection during the planning stage of the project lifecycle and using only subjective data from experts via questionnaires. Contractors can have a long-lasting effect on a single project. An investigation of contractor performance throughout the project lifecycle against sustainability success criteria is a good avenue for future research. For data collection, a sustainability performance questionnaire could be created and distributed at regular intervals and stages of the project lifecycle combined with available empirical measurements of contractor performance. For inference, a hybrid of FL for handling uncertain data and a machine learning model, such as the ANN, for making predictions and determining the trends in performance throughout the project lifecycle. This could lead to improved project and contract work efficiency and a potential metric for ongoing sustainable contractor evaluation throughout the whole project lifecycle.

### 5.2. Design Optimization

AI-based DSSs for design optimization in construction appears to be a new area of study with all research being published from 2018. There has been some work investigating sustainability goals; however, the benefits of different and hybrid AI models have not been fully explored yet. The primary stream for DSSs in design optimization is the optimization of material selection choices for sustainability. This has had minimal AI use for inference engine design until now, as there is only a single study recorded to have used AI for sustainable material selection. For future research, studying the benefits of multiple models of AI or combinations for material selection would be a fruitful avenue to pursue. In addition, consider other steps of the decision-making process during design. Ref. [52] utilizes RL for decision making in concept design; this is only effective in this study due to the highly similar nature of the designs being produced. With a dataset covering multiple design projects, a supervised learning approach, such as the use of an ANN [33], could be used for optimizing concept design for sustainability.

### 5.3. Dynamic Performance Prediction

There is an increasing trend towards continuous performance prediction throughout the project lifecycle. At present, there have been studies that consider AI models and

others with sustainability criteria, but a study focused on the continuous measurement of performance against sustainability criteria using an AI inference engine is an avenue to be pursued. This can be investigated with the intention of determining the readiness for this transition in the construction sector or the development of a framework to achieve this. An example of an approach that could be taken for this would be [54], who developed a six-stage guideline for applied machine learning in construction. It starts with problem definition and data selection and then data preparation and pre-processing, training the baseline estimator, creating interpretable machine learning, training the final estimator, and deployment and scoring. This is a comprehensive guide that could be applied for predicting a variety of performance metrics. A challenge related to this avenue of research would be for the collection and verification of newly defined data at regular intervals related to sustainability criteria over multiple construction projects. Furthermore, the complexity of the inference engine would need to accommodate data for prediction, which can change throughout all stages of the project lifecycle and work with incomplete and both linguistic and empirical data.

### *5.4. Safety Risk Assessment*

Another future direction would expand on the SRA field of study. The use of near-miss reporting as a dataset, considering circumstantial information alongside the quantitative methodology stated in [22,43,75], may be an avenue to pursue. This may allow for other forms of AI to be utilized. Using near-miss reporting, according to the Heinrich accident triangle [76], there are approximately 300 near misses for every 30 minor accidents and 1 major accidents. This would be a much larger pool of data for determining trends in accidents at work. This also allows a company to pre-emptively reduce accidents through analysis of near miss reporting. Challenges to this may be in the difficulty in determining an accurate metric for showing the improvement made by the system.

### *5.5. Site Logistics*

Improving the efficiency of site logistics has shown to improve the sustainability of projects, as discussed in Section 4.1.6, but there is opportunity for further study in utilizing AI models for the decision-making process. Ref. [71] developed a system that looked at the material transport routes, the emission levels from vehicles, and the size of the transport vehicles but relied heavily on the experience of experts. This produced only 47 alternatives to choose from for increasing sustainability. Using this same approach but incorporating the predictive capability of machine learning algorithms such as ANN would optimize the resultant transport routes. The nature of training a neural network would lead to the consideration of a significantly larger pool of alternatives to compare from and objectively identify the optimal course of action to take. Studying the benefits of AI-based DSSs for material supply for improving logistic sustainability can be seen as a good focus for future research.

## 6. Conclusions

This study set out with the aim of determining the current situation and trends for research into both the use of AI and for the consideration of sustainability in the development of DSSs in the construction project lifecycle. This study shows in Figure 4 that DSS research in the construction sector has received increasing attention over the last decade and that all studies could be broken down into six categories for further analysis. Most studies (50%) focused on EPP with DPP (15%) receiving the second-most attention, then there was supplier and contractor evaluation (12%), and finally SRA (6%), design optimization (6%), and site logistics (6%). EPP receiving the most attention is understandable, as the impact of EPP is crucial at the initiation and planning stages of a project. The decisions can lead to the selection or deselection of projects [77] or have large impacts on budgeting [78]. The DPP applies similar methodologies to EPP but for

continuous assessment throughout the project lifecycle. This appears to be the direction in which research is going.

After a large quantity of studies have been conducted for project performance prediction at the early stages of the project lifecycle, the latter half of the decade saw an increase in DPP studies that focused on all areas of the project lifecycle. This increase in studies was not restricted only to a change in stage, but other tasks related to site logistics, design optimization, and SRA all took place in the latter half of the decade. This would suggest the research in the field for DSSs in construction are broadening in application as well as the methodologies used. From a perspective of applied AI, the type of AI used was dependent on the category of application. The nature of problems involving prediction with historical data are ideal for tools such as machine learning models, especially ANN, with dealing with complex and non-linear relationships [69]. The unfortunate aspect of 50% of studies focusing on EPP is that the quantitative results will have a bias towards the individual results for this application of decision making. From Table 2, the most used AI models were the ANN and FL, followed by GA. Overall, 46% of all studies adopted a hybrid of more than one AI algorithm. The evidence discussed in the study of these articles show the use of hybrids to reduce the disadvantages of individual AI algorithms.

FL has been used for all categories of DSS except design. This is a useful tool when the data are based off experts' opinions [31,47,48,79]. With the nature of construction projects relying heavily on the experience of personnel [2,3], it makes sense that one of the most widely used AI models is FL, as it is useful for quantifying linguistic data. As can be seen in Table 2, the goal of sustainability, which has accumulated the most interest, is the economic goal of sustainability, with 87% of all studies focused on this. Figure 5 shows a near-consistent number of research papers over the last 10 years, which all focus on the economic goal of sustainability. The environmental and social goals of sustainability were considered in 30% and 38% of all papers, respectively. Figure 5 shows a trend of increased interest in the environmental and social goals over the latter half of the decade, expressing an increase in interest for project sustainability. This agrees with [5], as there has been a push to focus on multiple sustainability factors from the start of this century, while economic results have always been a key consideration in project management.

The novelty presented in this article is an in-depth study of the current and future trends related to sustainability and AI in the development of DSSs for projects in the construction sector. This benefits the known body of knowledge by providing insight and examples of key areas for future research in this field as well as presenting evidence of trends related to both AI and sustainability in relation to projects in the construction sector. When it comes to research limitations, the study only focused on the years from 2010 to 2021; however, this study was conducted in 2021, so the data for this year are incomplete. The papers are restricted to those using the keywords stated in 3.1 for the three selected databases: *Scopus*, *Science Direct*, and *ProQuest*. Only peer-reviewed articles were considered and only articles in English.

**Author Contributions:** Conceptualization: C.J.S. and A.T.C.W.; methodology, C.J.S.; software, C.J.S.; validation, C.J.S. and A.T.C.W.; formal analysis, C.J.S.; investigation, C.J.S.; resources, C.J.S.; data curation, C.J.S.; writing—original draft preparation, C.J.S.; writing—review and editing, C.J.S. and A.T.C.W.; visualization, C.J.S.; supervision, A.T.C.W.; project administration, A.T.C.W. All authors have read and agreed to the published version of the manuscript.

**Funding:** This research received no external funding.

**Acknowledgments:** Craig Smith is funded by both the National Manufacturing Institute Scotland (NMIS) and the industrial partner Galliford Try Ltd. Particularly, the authors would like to thank Angus Ho Yin Liu from Galliford Try Ltd. for his support.

**Conflicts of Interest:** The authors declare no conflict of interest.

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
