# Peer review of "Advancements in Artificial Intelligence-Based Decision Support Systems for Improving Construction Project Sustainability: A Systematic Literature Review"

_informatics, doi:10.3390/informatics9020043_

Round 1
Reviewer 1 Report
- Some heading do not have an appropriate numbering system; see pages 4 and 5.
- Section 2.3 is very weak. While this section is one of the most important pillars of this research, the provided category is very weak. In addition, the authors didn’t use appropriate references for supporting different claims in this section.
- What is the aim of section 2.4. this section is very confusing and weak.
- I can not agree with The research methodology part. How could the authors find the paper? Keywords? In the title or abstract or all of them? It is not likely the result of Scopus is 624 while the result of science direct could be “1494” . Please search for the reason. In addition, when a huge number of papers are excluded from the obtained papers, it means that the research has been designed inappropriately.
- While many of the discussed papers in this study are not DSSs. Some of the are optimization,…. Why do the authors mention the DSSs in the title of the paper.
- How the authors review the content of the papers? Any coding techniques for reviewing the content of the obtained paper.
All in all, I can not suggest this paper at all. I listed just some comments while there are many more comments, and it seems some things have been joined in this research.
Reviewer 2 Report
The article is devoted to an overview of the current state of research on artificial intelligence-based decision support systems to improve the sustainability of construction projects. The literature is reviewed systematically to explore the use of artificial intelligence in the life cycle of a construction project along with consideration of economic, environmental, and social sustainability goals. 2688 research papers were reviewed, 77 articles were further analyzed, and the main tasks of decision support systems were classified. The results of the review showed that the mainstream of research is devoted to predicting projects at an early stage (50% of all articles), with artificial neural networks and fuzzy logic being the most popular artificial intelligence algorithms used. Hybrid models were used in 46% of all studies. The goal of economic sustainability is the most discussed in research: 87% of all works consider this goal.
Despite the satisfactory quality of the article, some shortcomings need to be corrected.
- The methodology of the review should be described in more detail.
- The inclusion and exclusion criteria should be defined.
- The number of analyzed papers is different in the abstract (77) and text (91).
- It is not clear why the authors reviewed 2688 papers to exclude 2597. Maybe, the inclusion criterion should be defined more clearly.
- The discussion section should be included with the discussion of findings.
- The novelty of the paper should be defined.
In summarizing my comments I recommend that the manuscript is accepted after major revision, including improving the methodology of reviewing.
Reviewer 3 Report
In this literature review paper the authors analyze the population of papers published between 2010 and the first quarter of 2021 to summarize recent development in application of modern computational techniques to decisions related with construction projects with the focus on sustainability aspects (economic, environmental, and social).
The paper is well organized, with a concise and clear introduction, definitions of basic notions important for the analysis, and informative description of the methodology. The analysis of the final sample of papers was divided in four parts:
Section 4.1 is devoted to the objectives of DSS. The authors divide the papers into six categories according to the decision problem. Each category is explained, and example papers are analyzed in detail to draw conclusions on common features of papers that belong to a category. This part is, in my opinion, the most interesting.
Section 4.2. analyzes the papers according to the types of algorithms applied to solve problems. In my opinion, the classification of algorithms is unclear. The first sentence of the section lists all categories (applied also in Table 2 that summarizes classification of the papers). This sentence is too long to be understandable. The algorithm types are not classified at the same level of detail – sometimes the focus is on the problem/application, sometimes – on the methods of approaching the problem. For instance, convolutional neural networks are a sub-type of artificial neural networks; classification and clustering can be done using a number of algorithms; fuzzy logic can be used also within ANN and, per se, is not a class of AI; natural language processing is a field in machine learning and can be done by means of many algorithms (SVM, Bayesian Networks, ANN, …). Thus, I claim this part of the analysis lacks scientific rigor – I could do with some arguments for the classification you use in this section. Another problem is the high concentration of abbreviations throughout this part of the text. It makes it extremely hard to read. What is CSB in Line 474? The last sentence (line 488-491) that summarizes findings is, in its first part, obvious (as selection of the tool depends on the type of the problem), and in its second part, related with the topic of Section 4.1.
I have no such reservations to Section 4.3 – here, the classification is unequivocal (economic, environmental, and social aspects of sustainability). Similarly, 4.4 uses a well-defined classification of life cycle stages of built assets. However, what is “initial stages of construction” in Line 525? The sentences in lines 525-529 and lines 538-542 are unclear.
The vistas for future research presented in Section 5 are in fact the summary and conclusions. I find them interesting and well justified. However, Section 6 seems unnecessary, as it partly repeats the findings, but in a reversed order. Please consider combining sections 5 and 6 in a single section Summary and Conclusions.
Reviewer 4 Report
The major issue that this reader has with the paper is its motivation: are the trends that are found in the thorough and commendable research performed by the authors in any way useful for the field of reference?
I usually associate trend detection with business consultants whose aim is to understand where it is more convenient to invest in the future (in terms of money, manpower, communication), but does such an activity constitute a scientific advancement in the field? Despite all its clarity in formulating research questions, describing methodologies, and presenting results, the papers failed at providing a satisfying answer.
Apart from this (perhaps ideological and idiosyncratic) problem, some rewriting is required in what follows:
Section 2.3 appears to be unjustifiedly imbalanced: why are details provided to a certain level for Machine Learning, whereas other endeavours are left with no more than a very synthetic sentence? One would think that ML plays a more important role than the other endeavours, but given that in the abstract the primary role of Fuzzy Logic is highlighted, this unequal distribution of attention has no apparent explanation.
Section 2.4. is very synthetic, too synthetic: the reader is left wondering what kind of projects the article is focusing on. The authors call them “construction projects”, but are they really about construction in terms of physical buildings? Or is “construction” here used metaphorically, to refer to something other field? The project lifecycle description remains at such a general (even generic?) level that the reader is none the wiser about the context at the end of Section 2.4.
Round 2
Reviewer 1 Report
Dear Authors, I strongly believe the most important section of each literature review is its research methodology. When a research methodology has been designed wrongly, definitely its results are not reliable. As an expert in this area, I strongly do believe that section 3.1 is under question, and using these keywords many papers are overlooked and missed in this study. So, the results are not reliable. I can at least 200 papers in this research area that can not be identified using this strategy.
Reviewer 2 Report
Thanks for the authors for considering most of the comments and recommendations. In my opinion, now paper can be accepted.
Author Response
Please see the attachment.

This manuscript is a resubmission of an earlier submission. The following is a list of the peer review reports and author responses from that submission.
Round 1
Reviewer 1 Report
In this paper, the authors present a review of advancements in artificial intelligence-based decision making for improving construction project sustainability. The paper is well written and nicely structured but there are still subjects to addressing the following minor-moderate issues in order to increase quality and impact:
• The conclusion is too long, I suggest for the authors split it into two sections: Conclusion, Future Research and Recommendations.
• For the high-quality work, centralize, justify figures and tables and resize them into uniform size.
• References 5, 7, 31, 32, 33, there is a new line spacing at the end of these references, please correct it and do the same for similar mistakes.
• Some of the recent work in the area have not been mentioned in the manuscript. I, therefore, recommend having a look at these references and adding them to the paper:
- Review of deep learning: Concepts, CNN architectures, challenges, applications, future directions
- Optimizing the performance of breast cancer classification by employing the same domain transfer learning from hybrid deep convolutional neural network model
Reviewer 2 Report
It is a fine, useful paper. I think that you should mention the 61 papers that you have studied, as you say in Figure 1, in a separate table.
Reviewer 3 Report
I strongly believe, in a literature review, defining the scope of the review by using appropriate keywords, and designing a research methodology are important. I do believe authors failed to consider these key factors in their research. There are some critical keywords in this paper, but it seems the authors do not pay enough attention to them. For example, I believe there are four main keywords including “Artificial Intelligence”, “sustainability”, “DSS” and “construction”. First of all, authors should define the scope of each keyword. What methods are included in AI? What is sustainability? As you know sustainability is an extremely broad area. And what is DSS? In some parts of the paper, authors said DSS and in some sections they said “decision making” like in the title. These are not the same. I suggest authors read the following books:
- Burstein, Frada, and Clyde W. Holsapple, eds. Handbook on decision support systems 2: variations. Springer Science & Business Media, 2008.
- Power, Daniel J. Decision support systems: concepts and resources for managers. Greenwood Publishing Group, 2002.
- Kersten, Gregory E., Zbigniew Mikolajuk, and Anthony Gar-On Yeh. Decision support systems for sustainable development: a resource book of methods and applications. Springer Science & Business Media, 2000.
In addition, I refer authors to some good papers that classified AI methods:
- Mouloodi, Saeed, et al. "What can artificial intelligence and machine learning tell us? A review of applications to equine biomechanical research." Journal of the Mechanical Behavior of Biomedical Materials (2021): 104728.
But the most problem of this paper is its research method. I believe this research method cannot guarantee to find all potential papers which are within the scope of this paper. The authors stated they used ScienceDirect, it means they probably missed many papers in other publishers, in addition as far as I know even the advanced search of ScienceDirect can not guarantee a systematic way to find relevant papers. The most frequent used database for literature review papers are WOS and Scopus. But authors didn’t use these data base. In addition, some keywords have been defined in the search process, while I do believe these keywords cannot cover all relevant papers and other synonyms should be incorporated in the search.
Lastly, analysing the data in this research is not correct. For analysing some special approaches are used to find key factors. I refer authors to some good papers that have a good research method.
- Kabirifar, Kamyar, et al. "Construction and demolition waste management contributing factors coupled with reduce, reuse, and recycle strategies for effective waste management: A review." Journal of Cleaner Production 263 (2020): 121265.
- Zhang, Qian, Bee Lan Oo, and Benson Teck Heng Lim. "Drivers, motivations, and barriers to the implementation of corporate social responsibility practices by construction enterprises: A review." Journal of cleaner production 210 (2019): 563-584.
Since the fundamental of this review paper is under question, I must reject it.